# OpenReview forum: "SteerDiff: Steering towards Safe Text-To-Image Diffusion Models"
_ICLR.cc/2025/Conference — ICLR 2025 Conference Withdrawn Submission_

### Official Review · Reviewer_ymyV · 2024-11-03

**Soundness:** 2
**Presentation:** 3
**Contribution:** 3
**Rating:** 5
**Confidence:** 3

**Summary:**

This paper introduces SteerDiff, a lightweight adapter module designed to enhance the safety of Text-to-Image (T2I) diffusion models. SteerDiff operates by identifying and modifying inappropriate concepts within the text embedding space, ensuring that the generated images comply with ethical and safety standards without compromising the model’s usability or image quality. Unlike existing safety measures that rely on text classifiers or require extensive fine-tuning, SteerDiff serves as an intermediary between user input and the diffusion model, offering greater efficiency and flexibility. Extensive experiments demonstrate that SteerDiff outperforms state-of-the-art methods in reducing the generation of inappropriate content and shows robust defense against various red-teaming attacks. Additionally, SteerDiff proves effective in concept forgetting tasks, highlighting its versatility for different applications.

**Strengths:**

- This paper proposes SteerDiff, a lightweight adapter module that serves as an intermediary between user input and the diffusion model. It effectively identifies and manipulates inappropriate concepts within the text embedding space, thereby guiding the model to generate images that comply with ethical and safety standards.

- Under various adversarial attacks, such as the white-box attack P4D and the black-box attack SneakyPrompt, SteerDiff demonstrates significant robustness, successfully reducing the probability of generating inappropriate content.

- Through extensive experiments on multiple datasets, including I2P and COCO-30K, SteerDiff excels in reducing the generation of inappropriate content while maintaining a low attack success rate against different attack methods.

**Weaknesses:**

- The paper primarily focuses on unlearning harmful content. Have you considered attempting to unlearn style and copyright aspects as well? I noticed that ESD includes such experiments.

- The paper was mainly conducted on SD1.4. Have you tried applying your approach to SDXL?

- The baselines compared in the paper are somewhat limited. There are other unlearning methods available, such as SPM [1], AdvUnlearn [2], and RECE [3], that could be included for a more comprehensive comparison.

- Additionally, it might be beneficial to incorporate some human evaluations. Human assessment can help measure not only the reduction of harmful content but also the quality of the generated images.

**Reference:**

[1] One-dimensional Adapter to Rule Them All: Concepts, Diffusion Models and Erasing Applications. CVPR'24
[2] Defensive Unlearning with Adversarial Training for Robust Concept Erasure in Diffusion Models
[3] Reliable and Efficient Concept Erasure of Text-to-Image Diffusion Models. ECCV'24

**Questions:**

See Weaknesses.

---

### Official Review · Reviewer_MWhZ · 2024-11-03

**Soundness:** 3
**Presentation:** 3
**Contribution:** 2
**Rating:** 5
**Confidence:** 4

**Summary:**

This paper introduces SteerDiff, an intermediary module designed to prevent text-to-image (T2I) diffusion models from generating inappropriate content. Through construct prompt pairs related to the unsafe concept and train models on these pairs, SteerDiff can identify undsafe concepts and steer them to the safe region. The experimental results demonstrate that the SteerDiff outperforms concept removal methods, like Erased Stable Diffusion and Safe Latent Diffusion, which could prevent inappropriate content while maintaining image quality and effectively defend against adversarial attacks.

**Strengths:**

1. The paper investigates an important and interesting problem: steering the Text-to-Image diffusion model away from generating unsafe contents.

2. The experimental results demonstrate the effectiveness of SteerDiff in filtering unsafe content while maintaining high image quality and accurately preserving the intended semantics of safe content.

3. The paper is well-written and easy to follow. It has good visualizations of the generated images by SteerDiff.

**Weaknesses:**

1. SteerDiff is quite similar to Latent Guard [1], as both methods rely on LLMs to construct unsafe-safe prompt pairs and train models to detect unsafe concepts in the embedding space. While SteerDiff’s motivation is to steer unsafe concepts into safe regions, this raises the question: is transforming malicious prompts genuinely better than outright rejecting them? The steering module could potentially introduce vulnerabilities, as malicious users might exploit the transformation mechanism to bypass safety filters.

2. More adversarial prompt attacks and defenses should be discussed, such as Ring-A-Bell [2],  MMA [3] and POSI [4].

3. Given that the model is trained on pre-selected unsafe concepts, how well can it generalize to unseen unsafe concepts? Would it be necessary to retrain the identifier and the linear transformation matrix W for effective handling of new unsafe concepts?

4. Are there any theoretical proofs supporting the idea that unsafe embeddings can be transformed into safe embeddings via a linear transformation? Additionally, as the steering parameter $\epsilon$ controls the intensity of the transformation, would increasing $\epsilon$ result in a decrease in the probability of generating inappropriate content? Furthermore, the use of a fixed steering parameter in SteerDiff may not account for the varying degrees of unsafe content in different prompts. Will relying on a model to learn the optimal steering degree offer more precise control?

[1] Liu, R., Khakzar, A., Gu, J., Chen, Q., Torr, P., & Pizzati, F. (2025). Latent guard: a safety framework for text-to-image generation. In European Conference on Computer Vision (pp. 93-109). Springer, Cham.

[2] Tsai, Yu-Lin, et al. "Ring-A-Bell! How Reliable are Concept Removal Methods For Diffusion Models?." *The Twelfth International Conference on Learning Representations*.

[3] Yang, Yijun, et al. "Mma-diffusion: Multimodal attack on diffusion models." *Proceedings of the IEEE/CVF Conference on Computer Vision and Pattern Recognition*. 2024.

[4] Wu, Zongyu, et al. "Universal prompt optimizer for safe text-to-image generation." *arXiv preprint arXiv:2402.10882* (2024).

**Questions:**

Please see the weaknesses above.

---

### Official Review · Reviewer_gnQT · 2024-11-04

**Soundness:** 2
**Presentation:** 2
**Contribution:** 2
**Rating:** 3
**Confidence:** 4

**Summary:**

This paper introduces a text embedding optimization method aimed at mitigating the generation of inappropriate content. Specifically, after giving the definition of unsafe concepts, unsafe terms and safe terms are obtained through some open-source datasets and LLM. A lightweight adaptor module (i.e., an MLP) is utilized to classify the safety of individual term embeddings. The embedding of inappropriate prompts is mapped to safe content through a linear transformation. In addition, the proposed method demonstrates the capability to remove inappropriate content and other specific concepts while also exhibiting resilience against red team attacks.

**Strengths:**

1. The SteerDiff method introduced in the paper does not require intensive computation and can effectively remove NSFW content and specific content.
2. Experiments demonstrate the effectiveness of the proposed method.

**Weaknesses:**

1. This paper lacks clarity in its presentation, leading to confusion in both the figures and other descriptions. For example, W is introduced in Equation 3 but specifically define in Equation 4; the description of Figure 1 fails to illustrate the significance of the blue blocks; in Figure 2 (a), the Text Encoder and Diffusion Model are placed in a trapezoidal frame, which can easily be misunderstood as the Unet; Figure 2(b) does not differentiate between blue and red points; the text embeddings are denoted as 'E' instead of the more standard 'e'; the lower part of Figure 3 lacks explanation.
2. The paper treats unsafe terms and safe terms as direct opposites, relying on the use of antonyms to ensure input safety. This approach risks altering the original semantics of the prompts, potentially leading to unintended consequences in the generated outputs.
3. The paper appears to primarily focus on optimizing the embeddings of prompts without providing adequate technical depth. The novelty of the proposed approach seems limited and fails to advance the understanding of the underlying mechanisms involved.

**Questions:**

1. How can we ensure that the LLM comprehensively covers all unsafe and safe terms? If the enumeration is not exhaustive, will this limit the classifier's generality, making it unable to recognize unfamiliar concepts?
2. Safe prompts can still lead to inappropriate content due to ambiguous wording. How does this method address that issue?
3. Since the LLM is involved in the training process for word-level modifications, why not allow the LLM to evaluate and rewrite the entire prompt directly? This should not significantly increase computational costs.
4. The paper does not specify which classifier is used to detect specific exposed parts. If NudeNet was selected, what is the rationale behind choosing only five metrics for evaluation?
5. Which version of ESD was utilized, ESD-u or ESD-x? Additionally, why are results from ESD on other NSFW detection tasks not provided ?
6. Given that prompts in COCO-30K are known to be safe, according to the paper's approach, these prompts should not be rewritten. Why does SteerDiff achieve better FID results than the baseline model SD1.4?

---

### Official Review · Reviewer_9mEP · 2024-11-04

**Soundness:** 1
**Presentation:** 2
**Contribution:** 2
**Rating:** 3
**Confidence:** 3

**Summary:**

This paper proposes a framework for removing unsafe content in the text encoder module of Stable Diffusion. This work generates a set of inappropriate concepts and trains an MLP classifier to identify them. Then, it uses vector shifts to steer the text embeddings away from these concepts.

**Strengths:**

The topic of this paper has practical significance.

**Weaknesses:**

1.	**The paper has poor writing and formatting quality.** The formula for stable diffusion in Sec. 2.1 is incorrect (this equation is unconditional diffusion, not Stable Diffusion), and Fig. 2 is unclear.

2.	Although the results presented in experiments look promising,**the technical contribution is limited.** Similar approaches for guiding models to generate images toward specific targets have already been introduced in prior works [1,2].

3.	**The experiments are insufficient.** It seems that the MLP identifier model plays an important role in the final results, but the paper lacks details on its structure or accuracy.

4.	In Sec. 5, Fig.5 is not sufficient to demonstrate the authors’ claim that “our approach can be effectively applied to selective concept forgetting without affecting other content”. **Quantitative results would be much stronger than examples.**

[1] Schramowski, Patrick, et al. "Safe latent diffusion: Mitigating inappropriate degeneration in diffusion models." Proceedings of the IEEE/CVF Conference on Computer Vision and Pattern Recognition. 2023.

[2] Yang, Yijun, et al. "Mma-diffusion: Multimodal attack on diffusion models." Proceedings of the IEEE/CVF Conference on Computer Vision and Pattern Recognition. 2024.

**Questions:**

Why use an MLP-based identifier? What's the insight behind it? How is its accuracy, and would fine-tuning on a larger model improve results?

---

### Note · Authors · 2024-11-15

I have read and agree with the venue's withdrawal policy on behalf of myself and my co-authors.